# Unravelling How Single-Stranded DNA Binding Protein Coordinates DNA Metabolism Using Single-Molecule Approaches

**DOI:** 10.3390/ijms24032806

**Published:** 2023-02-01

**Authors:** Longfu Xu, Matthew T. J. Halma, Gijs J. L. Wuite

**Affiliations:** Department of Physics and Astronomy and LaserLab, Vrije Universiteit Amsterdam, De Boelelaan 1081, 1081 HV Amsterdam, The Netherlands

**Keywords:** single-stranded DNA-binding proteins, single-molecule technique, DNA replication, DNA repair, DNA recombination

## Abstract

Single-stranded DNA-binding proteins (SSBs) play vital roles in DNA metabolism. Proteins of the SSB family exclusively and transiently bind to ssDNA, preventing the DNA double helix from re-annealing and maintaining genome integrity. In the meantime, they interact and coordinate with various proteins vital for DNA replication, recombination, and repair. Although SSB is essential for DNA metabolism, proteins of the SSB family have been long described as accessory players, primarily due to their unclear dynamics and mechanistic interaction with DNA and its partners. Recently-developed single-molecule tools, together with biochemical ensemble techniques and structural methods, have enhanced our understanding of the different coordination roles that SSB plays during DNA metabolism. In this review, we discuss how single-molecule assays, such as optical tweezers, magnetic tweezers, Förster resonance energy transfer, and their combinations, have advanced our understanding of the binding dynamics of SSBs to ssDNA and their interaction with other proteins partners. We highlight the central coordination role that the SSB protein plays by directly modulating other proteins’ activities, rather than as an accessory player. Many possible modes of SSB interaction with protein partners are discussed, which together provide a bigger picture of the interaction network shaped by SSB.

## 1. Introduction

Central to many genome-maintenance machineries are single-stranded DNA binding proteins (SSBs). These SSB proteins play a vital role in the maintenance of genomes by binding exclusively and transiently to ssDNA intermediates during DNA replication, recombination, and repair. By further interacting with different proteins crucial to all aspects of genome maintenance and recruiting them to their targets on DNA, the SSB protein plays a prominent role in bridging genome maintenance pathways and modulating their activity. Due to their interaction with DNA, they influence many other downstream processes, which include all the possible protein-mediated biological functions. Biochemical studies have demonstrated that SSB plays an essential role in DNA metabolism. However, the real-time interaction dynamics between SSB with DNA and its partner proteins have proven elusive owing to the limited averaged population and time resolution. The recent development of single-molecule assays, in combination with robust ensemble biochemical techniques and structural methods, have contributed significantly to our understanding of the molecular mechanisms of SSB, interaction dynamics with other protein partners, and the mechanistic interactions with partner proteins.

Here in this review, we examine their structural and genetic variety throughout the kingdoms of life, and focus largely on the most thoroughly studied SSBs, which suggest that SSB proteins from various organisms show several similarities, regardless of their considerable diversity. Structurally, SSBs consist of an oligonucleotide/oligosaccharide-binding (OB) fold and a flexible C-terminal tail. The OB-fold domain from the SSB binds with high affinity to ssDNA, and the C-terminal tail of SSB plays a crucial role in regulating the other protein partner activity. We covered the observables of interest for the single-molecule studies of SSB proteins, and investigated what we could learn from them. After binding and functioning on ssDNA, these SSB–ssDNA complexes need to be bypassed, dislodged, pushed, or reorganized along the ssDNA to complete replication, recombination, and repair.

This review emphasizes the function of the SSB protein as a central scaffolding protein, rather than an accessory player, contributing significantly to the storage and reliability of genomic information. In addition to their role in DNA replication, recombination, and repair, SSB proteins function actively in nucleating enzyme complexes that are crucial to genome biology. Altogether, the SSB proteins play a crucial, central, and as-yet underappreciated role in coordinating the biology of the cell under a wide variety of conditions. We provide a perspective into the future of single-molecule studies of SSB and open questions in the field, which include the extant question of how SSB interacts with cross-species proteins in the context of viral infection.

## 2. Classification of SSB

### 2.1. Properties of SSB

As suggested by the name, single-stranded DNA binding proteins bind to ssDNA. While this is mostly understood in the context of preventing re-annealing during lagging strand synthesis or during DNA damage repair, SSB proteins are vital in many other processes. All proteins of this broad class interact with single-stranded DNA, and some also interact with dsDNA [1,2,3,4,5,6,7].

### 2.2. Classification of SSB

As far as is known, all classified organisms with available genomes encode SSBs, suggesting that the role they play is essential to life processes at a fundamental level [8]. The role of SSBs is most saliently communicated as their role in replication, to prevent the re-annealing of single-stranded DNA, so that the template strand can be copied. The SSBs differ significantly from one another, and variations between different kingdoms of life trump intra-kingdom differences [9].

The SSBs follow several distinct architectures, of which we will survey a few. Often, we see a high degree of structural conservation of SSBs within a given family (Figure 1A), though this may not be reflected in sequence conservation (Enc 34 as an example [10,11]). Here, we divide SSBs by their kingdom of life, which corresponds to structural characteristics. We take several representative examples, for which there is single-molecule data. The three most studied SSBs at the single-molecule level are the prokaryotic *E. coli* SSB, the eukaryotic human RPA, and the viral T4 gp32 from bacteriophage T4 (Figure 1B).

The crystal structures of the mentioned proteins are shown in Figure 1B, and the domain organization of several SSBs is shown in Figure 1C. Several different DNA binding architectures exist; most bacterial SSBs are homotetrameric [12], including *H. pylori* [13], *Mycobacterium tuberculosis* [14], and *Mycobacterium smegmatis* [15]. For example, *E. coli* forms a tetramer to bind DNA [16], whereas human RPA is a heterotrimer [17], T4 gp32 is a monomer [18], and T7 gp2.5 is a dimer [19].

The oligomeric status of each SSB has important implications on binding kinetics, as proteins requiring oligomerization (or conformational changes in general) to bind will necessitate extra steps in the binding process, which will alter binding kinetics [20,21].

The oligomeric status of the different SSBs is one of the most important structural features, though they can also be differentiated by other important elements, such as the presence or absence of a C-terminal tail, which in most cases comprises an interaction interface with other proteins [20,22,23,24,25,26], and can inhibit DNA binding [22,24]. Other SSBs, such as the human mitochondrial SSB (mtSSB), lack a C-terminal tail [21], though human mtSSB can still interact with other proteins despite this [27]. The C-terminal tail also participates in oligomerization, as the *B. subtilis SSB* gene is similar to the *E. coli.* SSB, although it lacks a C-terminal tail and, consequently, the capability to form a tetramer, as the *E. coli.* SSB does [28]. While we do not entirely know the function of different C-terminal tails, it is an important structural factor in the differentiation of different SSBs.

**Figure 1 ijms-24-02806-f001:**
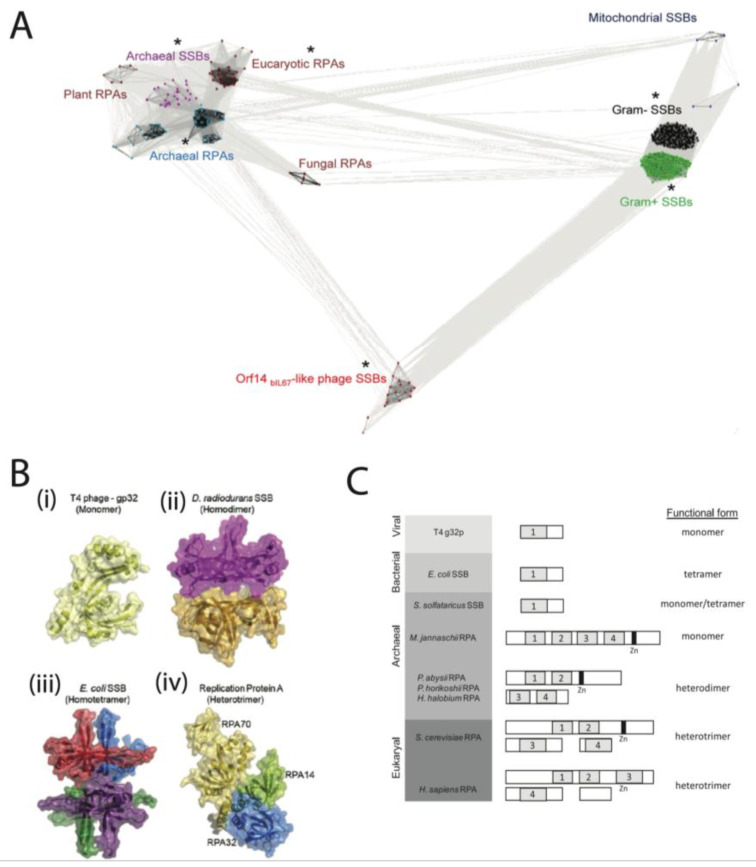
Classification and structural similarities of SSB proteins. (**A**) Cluster map of the SSB protein superfamily, as analyzed using CLuster ANalysis of Sequences (CLANS) [29]. In the protein network illustration, subfamilies that have members with known structures are indicated by an asterisk (*). The diagram represents individual proteins as dots, and phages encoding Orf14bIL67-like SSBs are highlighted in red. The rest of the proteins are categorized and coloured according to their origin: blue for Euryarchaea, maroon for Eukaryotes, purple for Crenarchaea, dark blue for mitochondria, black for Gram-negative bacteria, and green for Gram-positive bacteria. This representation allows for visual analysis of the distribution and diversity of SSBs across various species and kingdoms. Image from [9] under a Creative Commons Attribution License. (**B**) A graphical illustration of the crystal structures of single-stranded DNA-binding proteins from various organisms and their oligomeric states is presented. The structures are based on data obtained from the Protein Data Bank (PDB) IDs 1GPC, 3UDG, 1EYG, and 4GNX. Image from [30], with permission from Elsevier. (**C**) A diagram overview of the classification of SSBs within the three domains of life. The left panel shows a list of species arranged by domain. The middle panel shows an illustration of SSB and RPA monomers/multimers. The right panel depicts a functional subunit organization. Each numbered box represents one of the OB-fold domains. Images from [31], with permission from Elsevier.

## 3. Single-Molecule Toolbox to Study SSB

The investigation of SSB’s interaction with single-stranded DNA (ssDNA) and other protein partners is essential for comprehending cellular processes such as DNA repair, recombination, and replication. Despite extensive research, there are still numerous unresolved questions about the intricate stages of DNA metabolism that involve SSB proteins. Further studies into this subject matter hold the potential to provide crucial insights into these vital biological processes. The conventional bulk biochemical and structural methods used to investigate these processes often have limitations due to population averaging and are not able to address specific (mechanistic) questions. For instance, structural biological methods provide static pictures; hence, it is difficult to obtain the dynamic intermediate steps of a reaction. Over the last decades, a wide range of single-molecule techniques, such as optical tweezers [16,32], single-molecule Förster resonance energy transfer (smFRET) [33,34], magnetic tweezers [35], atomic force microscopy (AFM) and high-speed AFM [36,37], flow stretching [38,39], and nanopores [40], have been developed to study SSB proteins from various organisms.

### 3.1. Single-Molecule Force Studies of SSB–ssDNA Interactions

Generally, single-molecule techniques fall into two classes, namely those that measure force, displacement, and torque, and those that detect fluorescence. The first category, called single-molecule force spectroscopy, has become increasingly important for understanding the tensions, motions, and torques associated with biological molecules and their enzymatic activity. Studies have been conducted using single-molecule force spectroscopy to determine the interaction between different SSBs with dsDNA, ssDNA, or both [1,2,3,4,5,6,7], attempting to examine, for example, whether SSBs can destabilize duplex DNA. Among the various force spectroscopy techniques that can be used, optical tweezers are the most common one, primarily due to their ability to easily generate and manipulate ssDNA associated with SSBs and their feasibility of being combined with fluorescence microscopy.

With optical tweezers, DNA molecules have been manipulated to investigate the kinetics and thermodynamics of the binding of T7 SSB (gp2.5) and T4 SSB (gp32) to dsDNA and ssDNA [4,5,6,7]. An optical tweezer assay involves attaching one end of a DNA molecule to an optically trapped bead. On the other end, one of the following methods is employed: a micropipette ([41]), the surface of a microfluidic device ([42,43,44]), or a second optically trapped bead (as referenced in [45,46,47,48]) which is commonly referred to as a dual-trap optical tweezers setup. To study the effect of SSB on DNA molecules, double-stranded DNA is usually melted by force to obtain ssDNA [49]. Alternatively, SSB can be directly observed destabilizing duplex DNA. In the latter experiments, the dsDNA melting force was monitored in relation to the SSB concentration and pulling rate measured by elongation of end-to-end distance for the trapped DNA per time unit; Additionally, models were created to calculate the size of the SSB binding sites, referred to as the “footprint size”, as well as the association rates and equilibrium dissociation constants (K_D_) of SSB proteins binding to both single-stranded and double-stranded DNA [18,28,29]. Unlike other optical trapping approaches, dual-trap optical tweezers provide an advantage in that the DNA molecules are not fixed to a particular surface, allowing them to be moved between solutions using a multi-channel flow cell. This method provides a highly efficient method for probing the sequential interaction between SSB and other protein partners (See discussions in Section 5) and enabling the visualization of DNA–protein interactions with a minimum fluorescence background [50,51]. Moreover, the dual-trapping system is highly compatible with a variety of fluorescence imaging techniques that provide direct visualization of SSB binding dynamics to ssDNA, including wide-field [45,46,47,52], confocal microscopy [44] and super-resolution imaging (See Section 3.3 for more details). This compatibility is because a microscope objective is perpendicular to the DNA molecule, enabling straightforward imaging of interactions along its length.

Another commonly used assay to study SSB at the single-molecule level is the magnetic tweezer. In a magnetic tweezer assay, a biomolecule is tethered to a micron-sized superparamagnetic bead and a microchannel surface through antigen–antibody interactions. The corresponding force applied to biomolecules can be calibrated by analyzing the Brownian motion of the beads obtained through the bright-field images [53,54,55,56]. The relevant distance between the magnetic bead and the surface is determined by measuring the change in the diffraction pattern of the bead with respect to the magnet height [53,54,55,56] (Figure 2B). By varying the magnet strength and the experimental design, forces of between 0.001 and 100 pN can typically be achieved [54,55,57]. When compared with optical tweezers to study SSB, which is often limited by its lower throughput, magnetic tweezers allow many single DNA molecules to be tethered with separate beads and probed in parallel, thus, achieving high throughput of data collection. Example studies using magnetic tweezers were to determine whether the gp32 and *E. coli* SSB proteins could prevent DNA strand rezipping [2,3].

Other single-molecule force spectroscopy methods to study the SSB–DNA complex includes AFM, high-speed AFM [7,8], and nanopores [40]. The AFM approach represents a powerful means for imaging the ssDNA–SSB complex (Figure 2C) [58,59,60]. Advanced high-speed AFM is also applied to study the SSB, with an example application in visualizing the dynamics of SSB–DNA complexes in real-time [37]. This emerging AFM instrumentation allows one to observe the nanoscale dynamics of a system on a millisecond timescale while the sample is fully hydrated (reviewed in [61,62]). Nanopores can be used as a tool to investigate the binding affinity and selectivity of SSBs for ssDNA based on a distinctive electrical signature that is independent of either constituent in isolation [40].

**Figure 2 ijms-24-02806-f002:**
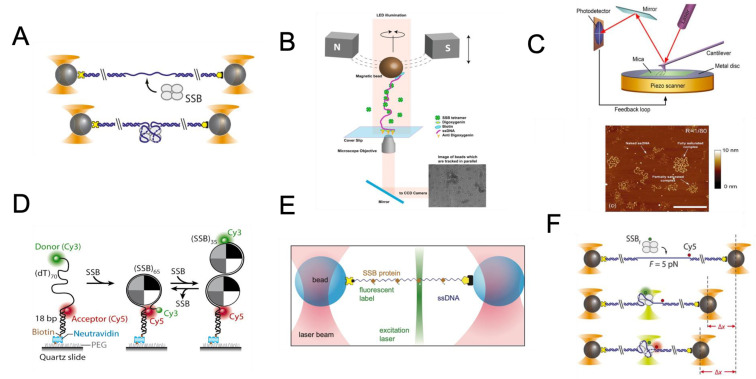
Overview of the Single-Molecule Toolbox for Studying SSB. (**A**) Dual optical tweezers for *E. coli* SSB unwrapping studies: This illustration shows a DNA construct containing two long double-stranded DNA handles, and a short single-stranded DNA fragment is trapped between two beads using optical tweezers. The binding of a single SSB tetramer is then studied by measuring the shortening of the DNA construct. (Adapted from Figure 1B of [16] with CC BY 4.0 License). (**B**) Magnetic Tweezers Assay: In this experiment, a unique DNA construct is secured between a cover glass and a paramagnetic bead, held in place by a magnetic field. The SSB-containing buffer is introduced and ejected to study the interaction between single-stranded DNA and SSB, providing insight into the mechanics of DNA-protein interactions. (**C**) Atomic Force Microscopy for Studying SSB-DNA Interactions: This illustration shows an AFM setup for imaging SSB and DNA interactions. The sample is probed using a flexible cantilever to obtain a three-dimensional image. An example image shows the different forms of M13 ssDNA-*E. coli* SSB complexes captured by AFM. (Adapted from Figure 2A of [36] under a Creative Commons Attribution Non-Commercial License (Copyright © 2007 The Author(s), http://creativecommons.org/licenses/by-nc/2.0/uk/, accessed on 30 April 2007). (**D**) Depiction of smFRET analysis for ssDNA wrapping mode determination by *E. coli* SSB. The binding of SSB tetramers in the (SSB)_65_ mode to ssDNA results in close proximity, yielding a high FRET value, while the (SSB)_35_ mode of binding leads to a lower FRET value. Adapted with permission from Ref. [63]. Copyright 2007 Elsevier. (**E**) Dual-trap plus confocal configuration: This illustration shows an example of fluorescently labelled T7 gp2.5 bound to biotinylated DNA held between two streptavidin-coated beads by trapping beams. The binding density of SSB can be monitored using confocal microscopy. (**F**) Dual-trap plus smFRET configuration: This illustration shows an example of a fluorescently labelled *E. coli* SSB wrapping DNA experiment. Following the binding of an AlexaFluor555-labeled SSB, both the DNA extension change and single-molecule FRET are measured simultaneously. Image adapted from Figure 4A of [16] under an Attribution 4.0 International (CC BY 4.0) License.

### 3.2. Image Measurement of SSB-ssDNA Complex

Another significant category of single-molecule tools is based on the detection of fluorescence. Several single-molecule fluorescence approaches have proven to be particularly useful for studying the SSB–DNA complex, for which smFRET has provided a high-resolution dynamic picture of how SSB interacts with ssDNA (Figure 2D). The smFRET technique involves the use of two fluorescent dyes, which are covalently attached to specific locations within the DNA molecule or its interacting protein. The smFRET assay can be performed using either confocal microscopy of freely diffusing molecules or TIRF microscopy of molecules attached to surfaces [64]. The smFRET measurements are frequently combined with other stretching techniques to provide a more comprehensive understanding of the DNA-protein interactions. When the distance between the two fluorophores is short (usually less than 10 nanometers), the donor transfers energy without radiation to the acceptor, resulting in the emission of fluorescence by the acceptor instead of the donor [65,66]. The FRET efficiency, defined as the efficiency of energy transfer from the donor to the acceptor, depends upon the proximity between the two fluorophores; therefore, it can be used to measure shifts in the distance up to ~10 nm. This tool is excellent for tracking real-time conformational and relative position changes within single biological molecules. The generated data, by measuring the dynamic states in a molecular system, can be quite different depending on the investigated system. Data fitting in smFRET data analysis is critical in understanding molecular dynamics and, thus, should be adopted based on the research question. Classic examples of smFRET to study the SSB–ssDNA complex are direct demonstrations of *E. coli* SSB in its (SSB)_65_ binding mode diffusing along ssDNA [67], which is consistent with early ensemble studies [68,69,70,71]. In addition to smFRET, fluorescence correlation spectroscopy (FCS) was also used to investigate the binding mode of SSB to ssDNA based on the detection of the hydrodynamic radius of SSB–ssDNA complexes [72].

### 3.3. Hybrid Single-Molecule Tools

Besides the independent use of single-molecule force spectroscopy and fluorescence microscopy, combined force manipulation and fluorescence visualization have been extensively exploited to probe the binding dynamics of SSB to ssDNA and its interaction with other protein partners (see Section 4 and Section 5, respectively). These combined approaches are instrumental in understanding, for example, DNA–binding protein interactions that are sequence-dependent [46], for monitoring protein translocation along DNA [52,73,74], and for examining the relationship between protein binding and the mechanical properties of DNA [7,32,75].

### 3.4. Example Output of Single-Molecule Studies

From single-molecule experiments, several parameters can be extracted (as summarized in Table 1, with detailed discussions provided in Section 4 and Section 5). For basic properties, there are the binding properties, which include the binding rate constants k_on_ and k_off_, corresponding to the on rates and off rates, respectively. In addition to binding properties, one can determine stoichiometries to gain insights into the binding footprint of an individual SSB binding event. Information on the kinetics of binding can also be obtained, such as the presence of one, two, or multistage binding. As well as measuring these parameters, one may also measure their dependence on experimental conditions, such as ionic concentration, template tension, temperature, and pH. One advantage of single-molecule experiments over bulk measurements is their capability to also measure the base sequence of the DNA template dependence of these parameters.

Once bound, an important value is the diffusion constant, which can determine if the SSB is stationary or diffusive (such as *E. coli* SSB [67]). We can also test cooperativity through the concentration-dependent binding kinetics. Lastly, single-molecule techniques also allow for the direct observation of interactions between SSBs and other proteins, such as the role of T7 gp2.5 in replisome coordination. These different parameters come together to describe the system of SSB interactions with DNA and with other proteins. For each parameter, there are well-described experimental and data analysis techniques in order to calculate the value. The single-molecule value obtained can recapitulate the values obtained by bulk measurement [79].

Additionally, we can calculate maximal coating densities, which can be used to calculate the binding footprint. These can be determined by finding an association between DNA length shortening and fluorescence intensity. This is typically linear, as one SSB will induce a near-constant contour length change by bending nucleotides in its OB fold or wrapping the DNA around a tetramer in the case of *E. coli* [30]. The shortening of the DNA will be directly correlated to the fluorescence intensity, which serves as a proxy for the number of SSB bound. When the system is saturated, one can calculate the total contour length change and divide it by the number of SSB bound (found via fluorescence intensity). This allows one to calculate the length change per bound SSB, which is an important quantity in understanding the binding mechanism. For example, the length change by SSBs that wrap DNA is higher than those that merely bend it within an OB-fold. This information can be compared with what is known from crystal structures. We know that the linear length of DNA is approximately 0.6 nm/nt under light tension. If the crystal structure of the SSB is complexed with DNA, it is possible to calculate the length change by finding the Euclidean distance between DNA bases on the 5′ and 3′ ends of the SSB and subtracting that from the expected distance of that number of nucleotides in a linear chain (0.6 nm/nt). 

Single-molecule experiments provide insights into the binding footprint of SSB (single-stranded DNA binding) proteins by analyzing the maximal occupancy, which is the point where an increase in fluorescent intensity stops, even with an increase in concentration along the bound DNA. By determining the number of SSB proteins, the average linear occlusion of DNA per bound SSB can be calculated by dividing the number of nucleotides by the number of SSBs. However, it should be noted that the average linear occlusion is not equivalent to the binding footprint, as some SSBs bind disorderedly. Interestingly, the SSB of phi 29 binds in a consistent manner, forming a “unit cell” with a nearly constant spacing of 3.4 ± 0.3 nucleotides per phi 29 SSB monomer [80]. The universality of this behaviour is unknown, as SSBs are usually thought to bind randomly. Based on this assumption and the one-dimensional parking problem [81], the maximal fractional occupancy can be calculated to be 74.8% on a DNA strand much longer than the length of a single SSB protein binding footprint [81].

Thermodynamic aspects of binding can be determined through bulk methods, such as isothermal titration calorimetry (ITC) or melting experiments, but they can also be probed by the single-molecule techniques. At a rough level, it is possible to determine the binding and unbinding as a function of force. Most proteins will be evicted from DNA held at high tension. Another glimpse into the thermodynamics and binding mode is by measuring the saturation dependence of certain binding parameters. Since salts shield the negative charges along the DNA backbone, along with certain amino acid sequences, such as the C-terminal tail of gp2.5 [23] as well as *E. coli* [22], information about the electrostatics of DNA–protein interactions can be garnered by varying the salt concentration of monovalent, bivalent, and polyvalent anions.

Monovalent anions are much less effective at shielding, even when normalized per unit charge than bi- or polyvalent anions. Bivalent anions pack double the charge in a more compact volume [82], allowing it to come close to the DNA or protein to screen the negative charge [83,84,85]. The change in binding properties as a function of mono-, bi-, and polyvalent anions may possibly be analyzed to determine the relevant length scales of the electrostatic interaction and the allosteric exclusion emerging from the close contact of the protein with DNA.

While not going into that level of sophistication, studies have gained insights into the ionic factors of DNA condensation and looping. For multivalent ions, those with centrally concentrated charges, such as Mg^2+^ and [Co(NH_3_)_6_]^3+^, result in lower persistence lengths than the polyamines putrescine^2+^ and spermidine^3+^, in which the charge is linearly distributed [86]. This observation, in addition to the preferential biding by [Co(NH_3_)_6_]^3+^ over spermidine^3+^ and the lower critical concentration of [Co(NH_3_)_6_]^3+^ for DNA condensation [87], provides insights into the condensation mechanics of DNA.

## 4. Examine the Interaction between ssDNA with SSB

### 4.1. General Binding Dynamics of SSB

SSB proteins play a critical role in binding ssDNA. The mode of SSB binding can vary, with some forming multimers, exhibiting cooperativity or exhibiting strong periodicity (as seen in phi29 [88]). Single-molecule experiments offer a deeper understanding of SSB behaviour, allowing the determination of binding and unbinding constants and the exploration of factors such as sequence dependence, DNA-conformation dependence, and conditions such as salt concentrations, temperature, pH, crowding agents, protein concentration, and the presence of co-factors. Additionally, single-molecule methods enable the investigation of multiple binding modes, which can be challenging to study in ensemble assays. Stoichiometries can provide insight into the binding footprint of SSBs and further our understanding of the structure of these proteins.

When bound, one can measure the diffusion and lifetime of the SSB. Diffusion can be characterized by the diffusion constant, but it is also helpful to determine if there is a directional bias to SSB motion. Interactions with other proteins can also be studied, such as colocalization, assisted binding, and the potential impact on the function of other enzymes. For example, T7 helicase and polymerase proceed much faster when SSB is present (see discussions in Section 5.1 and Section 5.2).

Next, one can study the binding cooperativity of SSB. In single-molecule studies, cooperativity can be determined by a McGhee von Hippel fit of the bound fraction of the protein as a function of concentration, which yields a sigmoidal graph [89]. Cooperativity has been demonstrated to depend on the intrinsically disordered tails of SSB proteins [22,78]. Most studies of cooperativity remain bulk studies [90,91], although single-molecule studies reveal evidence of cooperativity in Sulfobus solfataricus SSB [76], *E. coli* [30,92,93], and Thermus thermophilus [94].

### 4.2. Binding Dynamics of SSB to ssDNA under Tension

In addition to the general binding properties of SSB proteins discussed in Section 4.1, which can be probed with both bulk assay and single-molecule studies, the binding dynamics of SSBs to ssDNA under tension can be studied exquisitely with the single-molecule tools, such as by using optical tweezers [16,32] and magnetic tweezers [35]. The force-dependence of binding often depends on the binding mode, which varies between SSBs, from the monomeric binding in an OB-fold by T7 gp2.5 SSB [95] to the wrapping of DNA by *E. coli* SSB [30]. As *E. coli* SSB is highly sensitive to force, the unwrapping of the DNA from *E. coli* SSB begins at tensions as low as 1 pN, and complete dissociation occurs between 7 and 12 pN [33]. The impact of force is investigated in [16], revealing the tension-dependent wrapping behaviour of *E. coli* SSB (Figure 3A). Three distinct force regimes are observed, namely loading, wrapping, and protein removal (Figure 3(Ai). These can be observed in position traces of SSB (Figure 3(Aii)), as well as histograms (Figure 3(Aiii)). The same study found a dependence between the wrapping mode, as determined by the number of nucleotides that are interacting with the protein, and the force level, showing that as the applied tension increases, fewer nucleotides are wrapped around the protein surface, but this occurs in several stable modes, namely at 65nt for low force (<1 pN), 56 nt (at 1–5 pN), 35 nt (at 3–8 pN), and 17 nt (at 8–11 pN) (Figure 3B) [16]. These binding modes also proceed sequentially (Figure 3C) [32].

The binding dynamics for T7 gp2.5 have been investigated at different pulling timescales to investigate the prevention of secondary structure formation and the impact of T7 gp2.5 binding on the energetics of DNA stretching (Figure 3D). The experiment observed a clear shortening with the addition of T7 gp2.5, and by varying the speed, it could limit the number of SSBs binding [6]. It was determined that under the fast-pulling regime, fewer SSB bind, and the force relationship was similar to that of naked DNA (Figure 3D).

The real-time dynamics of SSB binding have also been investigated via high-speed AFM imaging [37]. The *E. coli* SSB binds, diffuses, and dissociates, and this is shown in real-time with AFM imaging (Figure 3E). In the emerging high-speed AFM technique, high-resolution images of the sample can be obtained in a fully hydrated state, thus, allowing millisecond-scale visualization of the nanoscale dynamics of the system. The buffer conditions, such as cation types, concentration, and pH, as well as the length of the substrate, can be varied in order to gain a better understanding of how environmental factors affect binding dynamics.

**Figure 3 ijms-24-02806-f003:**
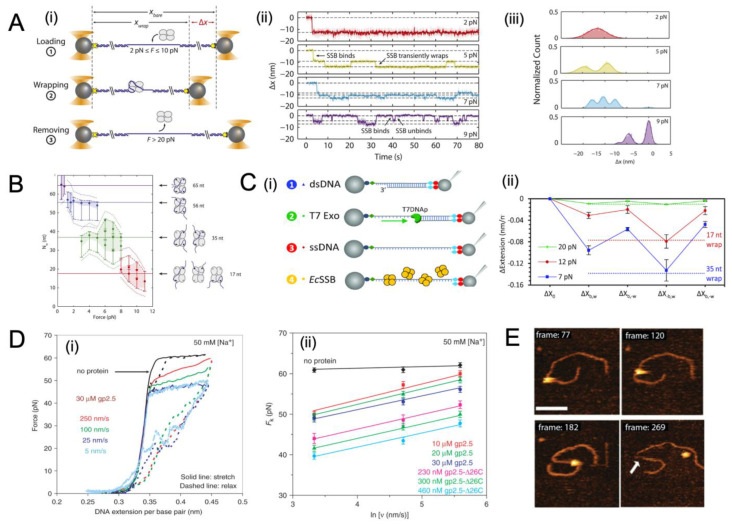
SSB interacting with ssDNA probed by single-molecule force spectroscopy. (**A**) Investi-gation of the intermediate states of ssDNA wrappers during SSB by single-molecule force spec-troscopy. (i) Schematic depiction of the SSB wrapping experiment under constant force. Between two optical traps, a DNA construct is held under a constant tension between 2 and 10 pN in the presence of proteins. The extension change is measured as a function of the SSB binding, wrapping, or unwrapping ssDNA. Following each observation, the SSB is removed by stretching the DNA construct to a high force (>20 pN). (ii) Representative time traces of SSB–ssDNA wrapping under 2, 5, 7, and 9 pN (with red, green, blue, and purple, respectively). The extension change data were collected at 66 kHz and averaged via a boxcar to 10 Hz (dark colour). All traces show SSB binding and compacting ssDNA as indicated by an extension decrease. With varying tensions, SSB exhibits a variety of intermediate wrapping states. The black dashed lines are indicative of the mean ex-tension change in each wrapping state. (iii) Distribution of extension changes for many SSB wrapping traces kept at constant tensions between 2 and 10 pN. This colour map corresponds to that in (ii). Solid lines represent multi-Gaussian fits to the distributions. Images adapted from Figure 2 of [16] under an Attribution 4.0 International (CC BY 4.0) License. (**B**) Plot of the number of wrapped nucleotides Nw as a function of various template tension F. The points are the best es-timates of Nw; dashed lines represent a tighter range of possible Nw values for each group of points derived from the SSB–ssDNA structure. Cartoon schematics depict possible wrapping modes cor-responding to the four groups. Rather than unwrapping gradually, ssDNA unwraps discretely under tension [32]. Images adapted from Figure 3 of [16] under an Attribution 4.0 International (CC BY 4.0) License. (**C**) The effect of tension on *E. coli* SSB–ssDNA binding dynamics. (i) Experimental method of determining *E. coli* SSB–ssDNA binding dynamics. A DNA construct of 8.1 kbp length is tethered between two functionalized beads (step 1, blue) with one bead held by a glass micropipette tip to extend the DNA, and the other bead held by a stationary optical trap to measures the template force. A long ssDNA molecule is produced by incubating dsDNA with T7 DNA polymerase (step 2, green) at 50 pN to trigger exonucleolysis to digest the bottom strand (step 3, red). Afterwards, ssDNA is held at a constant force and incubated with differing concentrations of *E. coli* SSB (step 4, yellow). At constant force, the DNA extension increases as a result of T7 polymerase strand diges-tion. Binding of *E. coli* SSB to ssDNA decreases DNA extension. (ii) The average extension decrease at each phase of binding for the *E. coli* SSB is presented under each force. It appears that following the free protein removal, SSB tetramers in the 17 nt (red dotted line) and 35 nt (blue dotted line) wrapped states are consistent with an average extension decrease at 12 and 7 pN across the ssDNA substrate, respectively. Images adapted from Figures 1A and 4B of [34], respectively under the terms of the Creative Commons CC BY license. (**D**) Experiment to investigate the effect of T7 SSB (gp2.5) and gp2.5-Δ26C on the DNA melting force in relation with pulling rate. (i) A pair of stretching (solid line) versus relaxation (dashed line) curves in the absence of protein (black) at a pulling rate of 250 nm/s, and in the presence of 30 mM gp2.5 at pull rates of 250 nm/s (red), 100 nm/s (green), 25 nm/s (blue), and 5 nm/s (light blue), respectively. (ii) The measurement of the non-equilibrium melting force, Fk (ν), as a function of pulling rate ν. Data are shown without protein (black diamond), and with 10 μM gp2.5 (red square), 20 μM gp2.5 (green triangle), 30 μM gp2.5 (blue circle), 230 nM gp2.5-Δ26C (pink square), 300 nM gp2.5-Δ26C (light green triangle), and 460 nM gp2.5-Δ26C (cyan circle). Linear fits are represented by continuous lines. In this study, data is collected in 10 mM Hepes (pH 7.5) and 50 mM Na+ (45 mM NaCl and 5 mM NaOH). Images adapted from Figures 1A and 2, respectively, of [6], under the Creative Commons CC-BY-NC license. (**E**) Direct visualization of the dynamics of SSB–DNA complexes using the high-speed AFM. Incubation with SSB was carried out on 69-gap-DNA substrates of different sizes under standard conditions (Tris-HCl, 50 mM NaCl, and 10 mM Mg2+). The images were acquired at a rate of more than one frame per second (720–990 ms). The bar is 50 nm. As indicated by the arrow, the exposed ssDNA region is visible after protein dissociation. Image adapted with permission from [37]. Copyright 2012, American Chem-ical Society.

### 4.3. Movement of SSB on ssDNA Probed with Single-Molecule Approaches

The distinction between diffusive and non-diffusive proteins is important. It is possible that diffusive proteins can cover a larger effective footprint (i.e., preventing secondary structure formation in this linear region of DNA). The SSB diffusion is passive and is thought to be driven largely by thermal motions. The diffusion of *E. coli* SSB has been observed in experiments using smFRET [67] (Figure 4). The DNA was labelled with donor and acceptor fluorophores located 69 nt apart, such that when the SSB was bound, there was a fluorescent signal produced (Figure 4A,B). It was found that there is free diffusion of the SSB along the DNA (Figure 4(Bii, top)). It is possible to ‘lock’ the *E. coli* SSB by forming a duplex structure with the bases at the 3′ and 5′ ends of the SSB–DNA complex (Figure 4(Bii, bottom)). In this case, the SSB does not diffuse. These observations are further demonstrated with a three-colour smFRET study with a longer ssDNA template (Figure 4(Biii, iv)). The findings showed that an SSB tetramer was capable of diffusing along the entire length of a (dT)130 ssDNA molecule [67].

Additional experiments were conducted to test two distinct diffusion modes of *E. coli* SSB, which differ by the relative motions of the SSB with the DNA (Figure 4C). The first mode, rolling (Figure 4(Ci–iii)), involves the DNA at the 5′ or 3′ ends lengthening or shortening by moving around the SSB tetramer. In this case, the relative position of a given DNA base and a given spot on the SSB tetramer do not move in relation to one another. The other diffusion model is that of sliding (Figure 4(Civ–vi)), where the ssDNA moves in relation to a fixed spot on the SSB tetramer. Experimental results support the sliding mechanism, as the site of the DNA FRET tag does not alter the FRET intensity pattern (Figure 4C).

Experiments have provided further insights into the mechanism of *E. coli* SSB diffusion and its association with wrapping mode [16]. Bulk studies observed multiple binding modes [96]. The single-molecule experiment simultaneously measured the position of DNA, as well as the point-to-point distance (via FRET) between a fixed point on the SSB tetramer and the DNA (Figure 4(Di)). They mapped the relationship between distance and FRET intensity (a measure of the relative distance between a fixed point on the DNA and the tetramer (Figure 4(Dii)), which were associated with four distinct binding states. The time series of FRET intensity versus position can be used to determine the transitions of binding modes from one to the other (Figure 4(Diii)) and estimated diffusion constants for each binding mode. The results show certain allowed transitions and support the idea of a linear kinetic pathway for wrapping [16].

**Figure 4 ijms-24-02806-f004:**
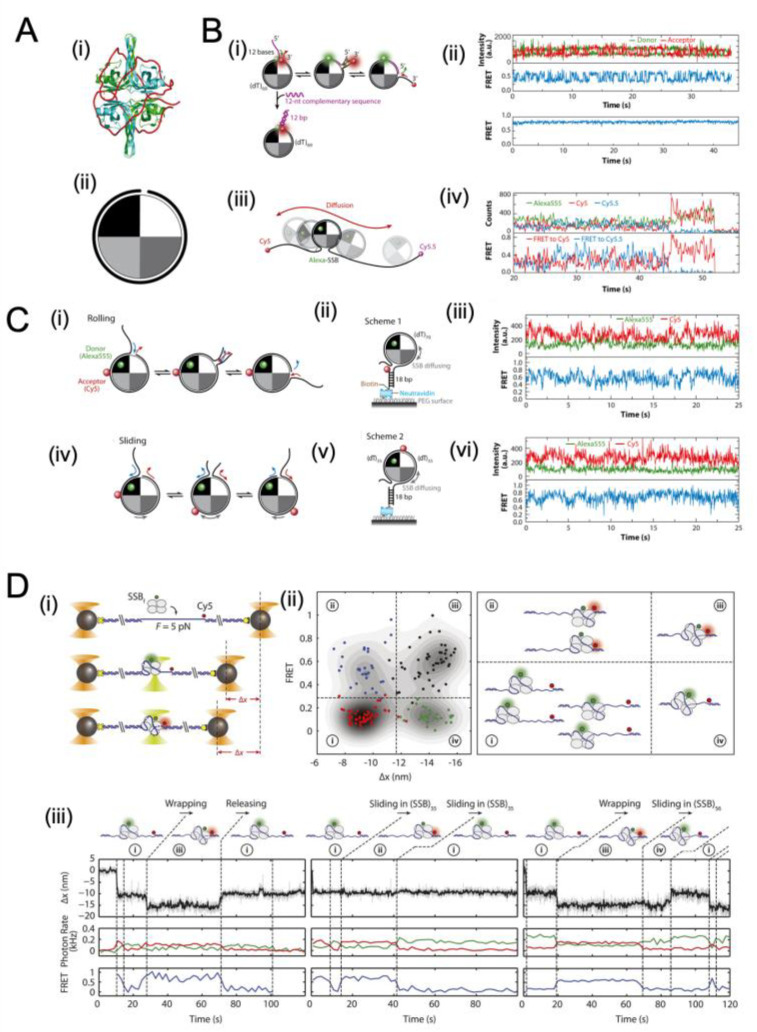
Single-molecule observations of SSB diffusion along ssDNA. (**A**) depicts an SSB tetramer bound to 65 nt of ssDNA based on a crystal structure [97] PDB ID of 1EYG. (**B**) demonstrates the diffusion of SSB along ssDNA using smFRET experiments. The results show rapid fluctuations in the FRET state due to SSB diffusion on 81-nt ssDNA (ii, top) but a reduction in fluctuations when ssDNA is limited to 69 nt (ii, bottom). The three-colour smFRET method was also used to demonstrate SSB diffusion along (dT)_130_ ssDNA(iii-iv). Images adapted from [67]. (**C**) depicts two hypotheses of SSB diffusion, the rolling and sliding mechanisms, with the latter being supported by single-molecule time traces. The results showed the same level of fluctuations in the donor and acceptor intensities regardless of the position of the acceptor on the DNA, suggesting the entire ssDNA sequence moves relative to the SSB protein surface. Reprinted with permission from Ref. [33]. Copyright 2011 Elsevier. (**D**) shows the binding modes and diffusion mechanism of SSBs, as measured by simultaneous fluorescence and DNA extension change, which provide information about SSB wrapping states and dynamics. Images adapted from Figure 4 of [16] under a Creative Commons Attribution 4.0 International license.

### 4.4. Sequence-Dependent Properties of SSB

For a nucleic acid binding protein, it is assumed that its interaction with the template is largely non-specific and that sections of the template are largely interchangeable and homogenous, except in the case of specific binding sequences, for example, such as Kozak sequences for translation initiation in eukaryotes [98]. While the assumption of a largely homogenous polymer may be useful in some applications, bases are often processed differently in important ways. Hairpins of GC-rich DNA require more force to unfold, but also have faster kinetics than AT-rich DNA [99]. GC-rich regions form a more stable secondary structure than AT-rich regions and more quickly. T4 gp32 and *E. coli* SSB proteins both act through the inhibition of refolding [2], so sequence manifests itself due to the different timescales of folding of GC-rich versus AT-rich hairpins. In the related case of mRNA translation, hairpins often slow and stall the ribosome, as the ribosome must resolve the secondary structure before proceeding [100], as the entry tunnel only allows ssRNA [101]. In the context of SSB, because most SSBs preferentially bind to single-stranded DNA, sequence dependence can manifest itself due to the higher likelihood of GC-rich DNA forming secondary structure. Such sequence-dependent mechanics has been probed using a template with large-scale spatial variation in GC content, as the resolution of the instrument is several hundred base pairs [51].

These above-described key observables include the diffusivity, binding footprint, binding characteristics and timescales, spatial heterogeneity in binding, and the existence of non-independent binding effects, such as cooperativity or recycling. These observables and derived quantities form a basis for the description of SSB and can be readily calculated from basic, standardized experiments (See Table 1). The impact of environmental factors, such as salt concentration and pH, on these parameters, can be assessed. The single-molecule approach enhances our understanding of SSB interactions with DNA and other proteins.

## 5. Coordination Role of SSB in DNA Metabolism

The recent development of single-molecule assays has contributed significantly to our understanding of the molecular mechanisms of SSB, interaction dynamics with other protein partners, and the mechanistic interactions with partner proteins. An overview of the interaction between SSB and other DNA binding partners in cellular activities is provided below, with a particular focus on how it interacts with replicative helicases, recombination repair helicases XPD and RecQ, replication restart helicases PriA, replication polymerases, recombinases, using DNA replication and DNA recombination as two major physiological processes of cells.

### 5.1. Overview of Single-Molecule Studies on SSB Interacting with Helicase

Helicases carry out many essential genome maintenance processes within the cell, such as replication, recombination, and repair [102,103,104,105,106]. It has been reported that the same helicase can carry out several of these functions [107,108,109]. Helicase activity must, therefore, undergo strict regulation. While there is no clear evidence regarding how this regulation occurs, growing evidence indicates that interactions with protein partners may be one of the mechanisms involved [105,107,108,109]. On the other hand, DNA helicases function to unwind dsDNA into ssDNA intermediate or to translocate along ssDNA, suggesting a frequent encounter with protein binding to ssDNA, such as SSBs, during various DNA processing events. Single-strand binding proteins have been shown to improve the unwinding efficiency of many helicases [110,111,112,113]. However, little is known about the consequences of encounters between translocating helicases and ssDNA-bound SSB. Here we review recent examples of single-molecule studies on the interaction between helicase and SSB, emphasizing replicative helicase, recombinational repair helicase RecQ and XPD, and replication restart helicase PriA protein.

#### 5.1.1. Interplay with Replicative Helicase CMG Complex

Replicative DNA helicases play an essential role in duplicating the genome in every cell cycle. Replicative DNA helicases are usually protein complexes with multi-subunit structures, such as the replicative helicase of eukaryotes, which is composed of 11 subunits and requires 2 subcomplexes and 1 protein to function. This heterohexameric helicase, the Cdc45-Mcm2-7-GINS (CMG) complex, is initiated through the formation of a complex with Cdc45 and the heterotetrameric GINS complex [38]. This CMG complex translocates in the direction of 3′–5′ along the leading-strand template and unwinds DNA at the replication fork powered by ATP hydrolysis [38,114]. In vitro single-molecule studies reveal that translocation on ssDNA of the yeast CMG helicase shows a rate at 5–10 bp s^−1^  [115], while the observed dsDNA unwinding rate to be 0.1–0.5 bp s^−1^, possibly slowed by a frequent long-lived pausing state [116,117]. Further studies of CMG-driven DNA unwinding with of ssDNA-binding protein RPA indicated that CMG complex translocates with a rate of ~8 bp s^−1^ at the fork [118], suggesting that the presence of RPA promotes the unwinding rates by CMG by 10–20-fold [38,118].

In a recent attempt [38] to directly visualize the interaction between unwinding CMG with RPA, fluorescently-labelled CMG complexes were monitored using forked linear dsDNA molecules containing a 40 nt polyT ssDNA (dT40) for CMG binding, and a Cy3 fluorophore for tracking translocation strand (Figure 5(Ai)). The fluorescent EGFP–RPA binds at the fork-terminal of the stretched DNA and colocalizes with the Cy3-labeled translocation strand. As a result, many 10 kb DNA molecules were completely unwound by CMG at an average rate of 4.5 ± 1.6 bp s^−1^. In agreement with recent single-molecule studies, they demonstrate that single CMG helicases are capable of unwinding thousands of base pairs of dsDNA with a rate comparable to that of ssDNA translocation by helicase [115,118]. The RPA-induced rate increase may be explained by the fact that RPA associates with the translocation strand behind CMG and obstructs helicase backtracking. The binding of RPA may also influence the activity of the helicase since it alters the interaction between the CMG and the excluded strand. The rate increase is significant compared to the stimulation associated with the unwinding of T4 gp41 helicase. The unwinding of the gp41 helicase from magnetically trapped hairpin templates has revealed occasional back slipping, but slippage was significantly inhibited when the T4 gp32 SSB was added [119,120]. It was also found that gp32 increased gp41 unwinding rates by 50% at low forces. Therefore, in that case, gp32 can promote the unwinding rate of gp41 by two mechanisms, namely by binding the translocated strand behind the helicase to prevent backtracking and by binding the excluded strand to assist in unwinding [38,118,120].

**Figure 5 ijms-24-02806-f005:**
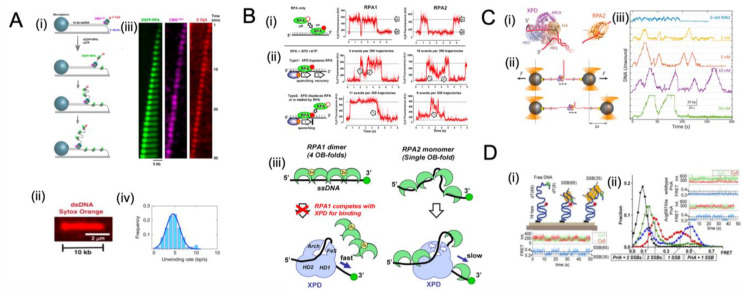
Overview of single-molecule studies on SSB interacting with helicase. (**A**) Direct visualization of the RPA-facilitated unwinding of processive forks by CMG helicases. The unwinding was performed on a 10kb λ-DNA fragment with a short fork construct (i), which was stained with Cy3 and immobilized on a cover glass. The LD655-labeled CMG was bound to the dT40 ssDNA in the presence of ATPγS. (ii) A representative snapshot of the flow-stretched λ-DNA stained with a fluorescent dsDNA intercalator, SYTOX Orange. EGFP-RPA and ATP were then introduced to initiate the unwinding, which was observed to occur as the EGFP-RPA bound both strands of the unwound DNA (iii). A histogram of the CMG-catalyzed DNA-unwinding rates is also shown (iv). Images adapted from Figure 1 of [38] under a Creative Commons Attribution 4.0 International license. (**B**) Two different mechanisms of xeroderma pigmentosum group D (XPD) helicase translocating along RPA-coated ssDNA were identified. Cy5-RPA binding to ssDNA was followed (i), and two distinct types of fluorescence trajectories were observed in the presence of XPD (ii). Type 1 showed a gradual decrease and increase in Cy5 intensity as XPD approached (arrow 1) and moved away (arrow 2) from Cy5-labeled RPA. Type 2 showed only a gradual quenching of Cy5-labelled RPA, yet no fluorescence recovery, indicating that XPD was displacing or stalling RPA. A model explaining how XPD targets RPA2-coated ssDNA and facilitates its translocation is also presented (iii). Images adapted from Figures 5 and 7 are reprinted with permission from [34]. Copyright 2012, American Chemical Society. (**C**) Single-molecule studies demonstrate that RPA2 enhances XPD helicase processivity. The processivity was studied using an optical tweezer assay to study hairpin unwinding by XPD in the presence of RPA2. Results showed that the processivity of XPD increased with RPA2 concentration. (**C**) Studies at the single-molecule level reveal that RPA2 enhances the efficiency of XPD helicase. (i) The left panel displays a schematic illustration of the FacXPD bound to a DNA fork, with roughly 10 nt embedded in the motor core and secondary contacts on each strand. The FacXPD design is based on the Sulfolobus acidocaldarius XPD structure (PDB 3CRV) [121]. The right panel shows a schematic representation of the FacRPA2, derived from a partial crystal structure of Methanococcus maripaludis RPA (PDB 2K5V). (ii) An optical tweezer assay was used to examine hairpin unwinding by XPD in the presence of RPA2. The increase in end-to-end extension of the hairpin construct, resulting from unwinding by XPD, was recorded and analyzed to determine the number of base pairs unwound. Arrows indicate the direction in which XPD moves along the single-stranded DNA. (iii) Representative traces of XPD molecules unwinding at a constant force in the presence of varying amounts of RPA2 (0–50 nM) demonstrate that XPD’s processivity increases with increasing RPA2 concentration. Images adapted from Figure 1 of [105] under an Attribution 4.0 International (CC BY 4.0) license. (**D**) A smFRET assay was used to investigate how PriA binds replication fork DNA in conjunction with SSB. (i) The SSB molecule transitioned between SSB_65_ and SSB_35_ binding modes, which were indicated by intensity fluctuation over time. (ii) Histograms of smFRET efficiency are shown for different combinations of DNA, SSB, PriA, and mutated PriA. Images adapted with permission from Ref. [122]. Copyright 2014 National Academy of Science.

#### 5.1.2. Interplay with Recombinational Repair Helicase XPD

Xeroderma pigmentosum group D (XPD) helicase belongs to subfamily 2B of helicases, including yeast Rad3 and human FANCJ, CHLR1, and RTEL [105,109,123,124,125], and is involved in a variety of DNA repair pathways. The XPD helicase mutation can affect nucleotide excision repair (NER) [126]. Human XPD is also associated with the transcription factor IIH and plays a significant role in the repair of nucleotide excisions [127,128,129,130]. Further evidence indicates that it also plays a role in chromosome segregation [131] and defence against retroviral infection [132]. While functioning on single-stranded DNA, XPD is likely to come into contact with other proteins, such as cognate SSB replication protein A (RPA). When the XPD encounters a bound RPA, it can bypass the RPA without dislodging it or facilitating its dissociation.

M. Honda et al. [34] reported a multi-colour single-molecule fluorescence method to simultaneously observe XPD translocating on ssDNA in the presence of cognate RPA. As a control, the binding and dissociation of RPA result in two-state fluctuations in the presence of XPD helicase (Figure 5(Bi)). After adding XPD and ATP, two distinct types of fluorescence trajectories were observed (Figure 5(Bii)). It was observed that the Type 1 trajectories were characterized by gradual plummeting of Cy5 events immediately followed by the gradual recovery of fluorescence, indicating XPD first approaches RPA and then bypasses it; however, in the Type 2 trajectories, only gradual Cy5 quenching was observed without recovery, suggesting that XPD may have displaced RPA or been blocked by bound RPA [34]. Furthermore, a comparison of the effects of both cognate RPA1 and RPA2 demonstrated that XPD translocation along ssDNA is affected differently by the presence of both RPA1 and RPA2. The statistical analysis indicates that XPD translocation is often accompanied by RPA1 dissociation actively facilitated by XPD, while XPD helicase does not displace RPA2 in most translocation events. It is important to note that the binding of RPA1 and RPA2 is different. The RPA1 extends ssDNA and strongly competes with XPD for binding, whereas RPA2 bends the DNA strand in preparation for XPD binding and slightly facilitates XPD binding. A reasonable assumption is that the binding mode of the SSB to ssDNA also has an effect on its fate in the event that a translocating helicase encounters it.

Along with single-molecule studies of RPA displacement by XPD, XPD unwinding activity in the presence of RPA was recently reported [105]. Previous work has shown that SSBs enhance DNA unwinding by SF2 helicases, but the mechanism by which this occurs is unknown [105,113,133,134,135]. B. Stekas et al. examined how the presence of RPA2 affects the XPD unwinding activity using a single-molecule optical tweezer assay [105]. They demonstrated that XPD repetitively engages in unwinding duplex DNA, while RPA2 increases this frequency with a high degree of processivity, or with a maximum number of base pairs unwound. Although RPA2 is capable of temporarily destabilizing duplex DNA, it does not promote XPD unwinding; rather, data suggest that XPD possesses a latent processivity switching mechanism regulated by RPA2. It is also important to note that no direct, specific interaction has been found between RPA2 and XPD in the solution [136]. These results may contribute to understanding how a protein that binds to ssDNA without any known protein–protein contact with the helicase can stimulate helicase-mediated DNA unwinding activity.

#### 5.1.3. Interplay with Recombinational Repair Helicase RecQ

In the case of recombinational repair, another well-studied example of helicase is RecQ. During recombinational repair, RecQ plays a role in repairing ssDNA gaps and dsDNA breaks in *E. coli* after the primary repair pathway, RecBCD, is inactivated [137]. It also has been demonstrated that RecQ is responsible for suppressing the production of illegitimate recombinants [138,139], resolving replication fork stalled events [140] and stimulating the SOS response in Escherichia coli. [141,142,143,144]. Additionally, *E. coli* defective in RecQ is susceptible to ultraviolet light, resulting in a decrease in the frequency of recombination that leads to impaired cell growth and death [142,145]. Single-molecule approaches have been used, complementary to bulk biochemical and structural tools, to analyze the interactions of single molecules with high spatial and temporal resolutions and reveal dynamic heterogeneities missing in ensemble experiments due to population averaging [146,147,148,149,150,151,152,153]. The following is an example of a single-molecule study [99], in which RecQ helicase molecules were in real-time tracked by using magnetic tweezers with or without a partner protein SSB. The results demonstrated that SSB dramatically increases the association and unwinding activities of wild-type RecQ, possibly stimulated by the C-terminal tail of SSB (SSB–Ct). However, further study shows that a mutant RecQ molecule lacking an SSB-Ct binding site can still be stimulated by the presence of SSB to a level of wild-type RecQ [144]. This stimulation could result from an unknown interaction between SSB and the helicase; alternatively, SSB may destabilize the dsDNA–ssDNA junction prior to the arrival of the helicase, promoting its movement forward [105,136].

#### 5.1.4. Interplay with Replication Restart Helicase PriA

The DNA replication protein complex can be dissociated before replication is completed by collisions with damaged DNA or immovable protein barriers [122,154,155,156,157]. Cells can resolve this potentially lethal problem by reloading the replisome by activating “replication restart” reactions [157]. In bacteria, the PriA DNA helicase orchestrates this vital activity by binding to structure-specific DNA and interacting with replication-associated SSBs [158,159,160]. Additionally, PriA also targets its activity to replication forks by interacting directly with SSB tetramers [122,134,161,162,163,164]. To better understand the intermediate steps of how PriA binds to replication fork DNA in conjunction with SSB and stimulates subsequent replication restart reactions, Bhattacharyya et al. [122] investigated, using a smFRET assay, the influence of direct interaction between PriA and (SSB) on SSB/DNA complex formation thereby exposing a potential ssDNA replisome reloading site. Without the PriA, *E. coli* SSB binds either 35 nucleotides per tetramer (SSB_35_) or 65 nucleotides per tetramer (SSB_65_), consistent with previous studies [63,96] (Figure 5(Di)). Upon addition of PriA, PriA demonstrates a more substantial stabilizing effect on the SSB_35_ binding mode than on the SSB_65_ mode and, thus, exposes more ssDNA owing to the SSB_65_-to-SSB_35_ transition. The FRET signal changes suggest that PriA binds to newly exposed ssDNA resulting in a slight reduction in the FRET efficiency state. The formation of a PriA-SSB complex has been suggested to result in structural alterations of the SSB/DNA complex, thereby exposing single-stranded DNA for capture by PriA. In a separate investigation, the PriC protein, which initiates an alternative replication restart pathway in *E. coli*, was demonstrated to preferentially stabilize the SSB_35_ mode of replication [165]. In conclusion, these results suggest that the remodelling of the SSB-binding mode may be a general requirement for DNA replication restart.

A summary of mechanisms for SSBs enhancing helicase activity can be classified into three groups [105,136]. Firstly, SSBs may be able to destabilize DNA duplexes at the junction of ssDNA–dsDNA prior to the helicase, thereby facilitating its movement forward. Alternatively, they could facilitate unwinding by stabilizing the excluded ssDNA strand and preventing helicase backsliding, thereby promoting the forward movement of the helicase. Lastly, they could stimulate the helicase to undergo processive unwinding by interacting directly with the helicase and the DNA complex.

### 5.2. SSB Interacting with Replicative DNA Polymerase during Primer Extension

Bulk studies have shown that the presence of SSB significantly enhances DNA replication in vitro [19,166,167,168,169,170]. This enhancement may be attributed to the multiple roles that SSBs perform (see review [19,30,171]), including the prevention of degradation of ssDNA, the removal of secondary structures, the increase in recognition and initiation of primers, a decrease in non-specific DNA polymerase binding to the template, and an increase in DNA polymerase’s activity in displacing strands and extending primers [19,30,166,167,168,169,170,171]. However, it remains unclear whether the seemingly conflicting roles of polymerase and SSB on ssDNA can be coordinated during the lagged strand replication.

Recent studies using single-molecule tools, such as optical tweezers and micropipettes, have attempted to determine the coordination between DNA polymerase and SSB, mimicking lagging strand synthesis [172] (Figure 6A). Real-time activities of two different DNA polymerases, namely human mitochondrial Polγ holoenzymes and bacteriophage T7 DNA polymerase, were measured with or without homologous and non-homologous SSBs. The instantaneous replication rate for both DNA polymerases increased rapidly within a force range where the secondary structure was likely to develop (<6 pN in their study case), suggesting that free DNA secondary structures pose significant challenges to the advancement of these replicative DNA polymerases. Furthermore, both polymerases demonstrated a maximum replication rate under the tension range where SSBs remained stably bound to templates (<8 pN in their study case). Conversely, both DNA polymerases failed to achieve their maximum replication rates when heterologous or mutant SSBs covered the DNA construct. These observations demonstrated that SSB binding causes destabilization of the secondary structure, which in turn favours maximum replication, but only in the case that a functional interaction between the replicative DNA polymerase and SSB is established that facilitates SSB release [172].

Similar increases in replication activity with homologous SSBs present were observed for the replication systems of mitochondria and bacteriophages [172], suggesting that polymerases (of these two organisms) are likely to employ similar mechanisms to displace firmly bound SSB proteins during the lagging strand synthesis (Figure 6(Aiii, iv)). Therefore, we assume that a higher polymerase/SSB interaction energy is needed to overcome the binding energy of SSB to ssDNA and, thus, release SSB from ssDNA due to the polymerase.

**Figure 6 ijms-24-02806-f006:**
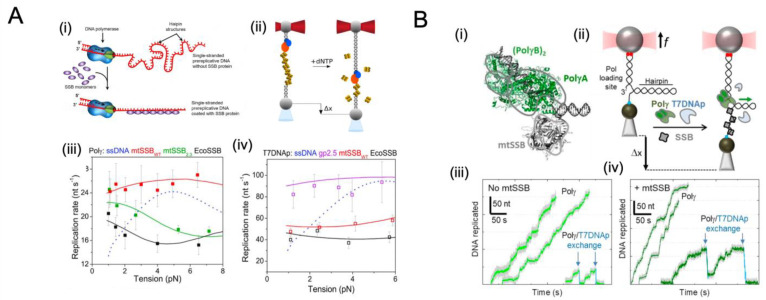
Single-molecule studies on SSB interacting with DNA polymerase during replication and strand exchange. (**A**) The effect of SSB proteins on DNA replication is shown in (i). SSBs bind to intermediate single-stranded DNA (ssDNA) during replication to prevent secondary structure formation and promote polymerization. However, the pre-bound SSB must be displaced for DNA polymerase to synthesize ssDNA into double-stranded DNA (dsDNA). Image adapted from [173]. The optical tweezer plus micropipette method used to measure replication is illustrated in (ii). The DNA polymerase binds to the primer template’s 3′ ends, while SSB coats the ssDNA portion. In the presence of deoxynucleoside triphosphates (dNTPs), polymerization causes SSB release, as seen by the extension of the DNA template. The maximum rate (Vmax, in nt s^−1^) of Polγ is shown in (iii) to be dependent on tension in the presence of various SSBs (mtSSB, *E. coli* SSB). Solid lines represent the best fit of the data with Equation (4) in [172] for quantifying the energetics of the polymerase– SSB interactions necessary for SSB displacement. As a reference, the dotted blue line indicates the fit to Vmax of the SSB-free ssDNA. The Vmax of T7 DNAp in the presence of different SSBs is depicted in (iv). Solid lines show the best fit of the data with Equation (4) in [172] to compare the homologous and non-homologous effects. As a reference, the dotted line shows the fit Vmax of T7DNAp in the absence of SSB. Images adapted from [172], under the terms of the Creative Commons Attribution Non-Commercial License (http://creativecommons.org/licenses/by-nc/4.0/, accessed on 17 December 2022), Copyright © The Author(s), 2019. (**B**) The interaction between Polγ (PDB 3IKM [174]) and mitochondrial SSB (PDB 3ULL [175]) during strand displacement synthesis is shown in (i). The optical tweezer plus micropipette method used to investigate strand displacement synthesis by Polγ and T7DNAp is demonstrated in (ii) [176,177] (iii) Representative traces of Polγ (2 nM) in the absence (left) and presence (right) of competing T7DNAp served as a control. (iv) Representative traces of Polγ (2 nM) activity in the presence of cognate mtSSB without (left) and with (right) competing T7DNAp. Images adapted from [177] under a CC-BY-NC-ND 4.0 International License.

### 5.3. Single-Molecule Studies on SSB with DNA Polymerase during Strand Exchange

The effect of SSB on strand displacement DNA replication has also been examined using single-molecule tools [176,177] (Figure 6B). It was previously demonstrated that strand displacement DNA synthesis, such as by Polγ, accomplishes replication by utilizing stable secondary structures [178,179], ensuring that the D-loop DNA structure is maintained at the origin of heavy strands [180], and removing primers through the coordination of primer processing factors [181,182,183]. However, the efficiency of Polγ is limited to a few nucleotides [182,184,185,186,187], in accordance with other DNA polymerases involved in strand displacement synthesis [176,188,189]. Bulk assays have shown that SSBs can stimulate the strand displacement replication of many other DNA polymerases [188,190,191,192,193], yet the effect of SSB on the strand displacement activity of Polγ remains unexplored. Recently, researchers have used single-molecule manipulation to quantify the effect of cognate and noncognate SSBs on the strand displacement mechanism of Polγ. Various concentrations of cognate mtSSB and noncognate phage T7 gp2.5 and *E. coli* SSB have been investigated to examine the potential role of species-specific polymerase–SSB interactions. The study demonstrated [176,177], in accordance with previous findings [188,190,191,192,193], that SSBs stimulate strand displacement DNA synthesis by utilizing a variety of mechanisms, including the binding of mtSSBs to displaced ssDNA to increase the destabilization energy and reducing the regression pressure on the holoenzyme. These stimulatory effects are also shown to be enhanced by species-specific functional interactions [176,177].

### 5.4. Single-Molecule Studies of SSB Interplay with Recombinase

During DNA metabolism, replication forks can stall or collapse, resulting in extensive single-strand gaps [194,195,196,197]. Consequently, the SSB protein binds to the ssDNA in these gaps, preventing other proteins from accessing the ssDNA. The RecA protein from *E. coli* is essential to repair broken DNA and maintain genomic integrity through homologous recombination. In order to function, RecA filaments are required to nucleate and grow on single-stranded DNA concurrently with SSB, which sequesters ssDNA continuously and thereby causes it to compete with and prevent RecA assembly [198,199]. Because of the complexity resulting from dynamic competition with SSB during self-assembly on ssDNA lattices, our knowledge of RecA filament assembly and its role in DNA recombination has been compromised. Despite extensive and varied efforts, ensemble measurements based on an averaged population are not able to distinguish between nucleation and growth in a reliable manner [198,199,200,201]. Several single-molecule assays have also been conducted to study the nucleation and growth of RecA on naked double-stranded DNA and ssDNA [202,203,204,205,206,207,208]. In spite of this, the general consensus is that once ssDNA has been generated by cellular metabolism, SSB attaches to ssDNA immediately before RecA nucleates and displaces the SSB.

A recent report from J. C. Bell et al. [39] used single-molecule approaches for measuring nucleation and growth rate on SSB-coated ssDNA while directly visualizing RecA filament assembly. Following the coating of non-fluorescent native SSB on ssDNA, the assembly of RecA filaments was investigated with a fluorescent RecA protein. With the evolution of the nascent clusters, they grew longer, and new clusters appeared. The data confirm previous findings that these mixed nucleoprotein complexes consist of rod-like clusters of RecA filaments sandwiched in between compact and flexible ssDNA coated with SSB (Figure 7A) [209]. Cluster formation increased linearly with time, and the nucleation rate of RecA showed a nonlinear dependence on RecA concentration (Figure 7(Aiii, iv)), which can be described by the equation J = k[RecA]^n^, where J represents the nucleation frequency, k represents a rate constant, and n is the number of protomers in a critical nucleus [203,210]. In the presence of ATP, the number of protomers in a critical nucleus is 2.2 ± 0.6 (mean ± s.e.m) (Figure 7(Aiv)). It is believed that nucleotide binding occurs in the binding pocket between monomers of RecA nucleoprotein [211], and prior findings support a model that dimers are the smallest oligomeric species capable of forming a stable nucleus on ssDNA. Earlier studies of RecA filament growth in terms of dsDNA and ssDNA, without SSB, have reported that RecA filaments grow in both directions [203,204]. Further observation of RecAf (green) growth from the preformed RecACy3 (red) clusters demonstrated that the growth of RecA filaments follows a bi-directional pattern, although it is greater in the 5′→3′ direction. 

It is believed that the recruitment of RecA to SSB-coated ssDNA and nucleation are the first steps in RecA-mediated DNA recombinational repair [39,212]. Once RecA is recruited, SSB is expected to no longer inhibit the extension of RecA filaments [200,213]. To further comprehend how RecA displace SSB, C. Joo et al. [204] used single-molecule FRET to observe the real-time dynamic interactions between RecA and SSB. Using a preassembled nucleation cluster and fluorescently labelled SSB-coated ss/dsDNA construct, they demonstrated direct evidence for the removal of SSB by an extending RecA filament. The FRET signal was observed to transition from low efficiency (with a preassembled nucleation cluster) to higher efficiency (with SSB coated on ssDNA) and back to low efficiency again (when SSB is displaced by RecA) (Figure 7B). In examining the single-molecule time traces of RecA replacing SSB, the rates of SSB removal and the rate of unhindered filament extension showed nearly identical values, indicating that SSB’s interference is minimal. The smFRET data also demonstrated that extending RecA filaments can efficiently push SSB along ssDNA at a rate equal to the growth of the filament, further confirming that bound SSB does not present a significant challenge [204].

As opposed to the above-described *E. coli* recombinase RecA, which forms nucleoprotein filaments rapidly in the presence of cognate SSB, Rad51 filaments are rarely observed to form or grow slowly in vitro when stimulated by the cognate RPA [208,214]. How RAD51 assembles into long homologous recombination-proficient filaments on an RPA-coated ssDNA in the presence of other recombination mediators, such as BRC-2 and RFS-1/RIP-1, remains unknown. In a study, O. Belan et al. [215] examined the mechanism of RAD51 filament growth on RPA-coated DNA by using single-molecule optical tweezers and confocal fluorescence microscopy. Unlike RecA, Rad51 nuclei in eukaryotes grow very slowly on their own. Therefore, the assembly of RAD-51 was initiated in the channels containing RAD-51 as well as the mediator protein (Figure 7(Ci)). It was found that substoichiometric concentrations of BRC-2 and RFS-1/RIP-1 had a significant effect on the assembly rate of RAD-51 filaments. This process was measured by observing eGFP fluorescence loss and parallel decreases in the force exerted on ssDNA caused by RAD-51 assembly and displacement of RPA-eGFP [216]. The RAD-51 growth stimulated by recombination factors was measured as a drop in force between the optical traps from ∼15 pN to ∼1 pN (Figure 7(Cii)), indicating an increase in the stiffness of RAD-51-coated ssDNA [216].

**Figure 7 ijms-24-02806-f007:**
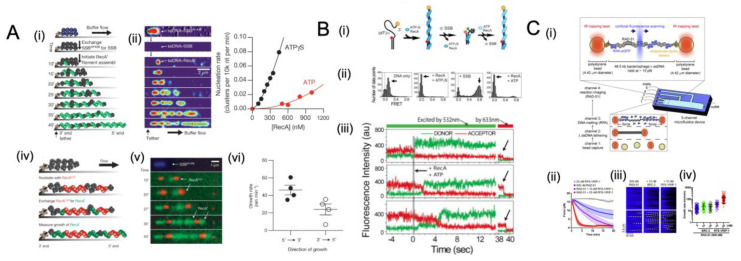
Examples of single-molecule studies of SSB interplay with recombinase. (**A**) Direct observation of RecA filament assembly on single-stranded DNA (ssDNA) coated with SSB demonstrates that RecA nucleates as a dimer and grows in a bidirectional manner. Visualization was done using TIRF microscopy in a microfluidic flow chamber. Results showed that higher RecA concentrations lead to increased nucleation rates and that the growth occurs faster from the 5′ end in the 5′→3′ direction with a rate of 44 ± 11 nm min^−1^. Images adapted from Figures 1 and 3 of [39]. (**B**) The displacement of SSB by RecA in the presence of a preformed nucleation cluster was shown using smFRET traces. Results indicate that RecA efficiently displaces SSB by quickly forming a filament. Reprinted with permission from Ref. [204]. Copyright 2006 Elsevier. (**C**) The assembly and growth of RAD-51 filaments on RPA-coated ssDNA were studied using optical tweezers and confocal fluorescence microscopy in vitro using fluorescently labelled *C. elegans* proteins. The RPA-eGFP fusion protein was coated on a force-melted ssDNA [49]. In the presence of ATP, the traps were moved to protein channels containing RAD-51 or mediator proteins to initiate the assembly of RAD-51. The assembly of the RAD-51 and displacement of the RPA-eGFP were followed by the loss of fluorescence of eGFP and the simultaneous decrease in the force exerted on the ssDNA associated with recombinase filament formation [216]. (ii) Forces are measured between the traps as a function of time in different RFS-1/RIP-1 concentrations; shaded areas represent SEM (n = 3 to 8 molecules). (iii) Representative kymographs of growing RAD-51 filaments (dark); the growth rate is expressed as the slope of the displaced RPA-eGFP signal. (iv) The quantification of growth rates under the indicated conditions. Images adapted from Figure 1 of [215] under the terms of the Creative Commons CC-BY license. Images adapted from Figure 1 of [215] under the terms of the Creative Commons CC-BY license.

### 5.5. Chemo-Mechanical Pushing of *E. coli* SSB by a Translocating Protein Partner

Both bulk and single-molecule assays have shown that SSB binds exclusively to ssDNA with very high (pM to fM) affinities [32,33,71,217]; however, these tightly bound complexes must be displaced, bypassed, or redistributed along ssDNA to complete replication, recombination, and repair. As discussed in Section 5.1.2 [34], a pre-bound RPA on ssDNA can be dislodged by a translocating helicase XPD or bypassed without dissociation. The PriA binding to *E. coli* SSB can modulate the binding mode from SSB_65_-to-SSB_35_ to expose more ssDNA for DNA replication restart. Considering the diffusive property of *E. coli* SSB, one other potential mechanism for reorganization can be pushed along ssDNA by a translocating protein. The following is a description of one such example (Figure 8).

The DNA translocases are motor proteins capable of translocating ssDNA at high rates powered by the hydrolysis of ATP [107,218,219]. To delve into the impact of a directional translocase encountering an *E. coli* SSB tetramer bound to single-stranded DNA, a smFRET assay was utilized to detect such pushing events [220] (Figure 8A). A fluctuating FRET signal is observed when Cy5-labeled *E. coli* SSB is bound to surface-immobilized 3′-Cy3–labelled ssDNA, indicating that SSB is randomly diffusing on ssDNA (Figure 8B). When adding Saccharomyces cerevisiae Pif1, a translocase for ssDNA 5′ to 3′, irregular-spaced saw-tooth FRET spikes are observed with ATP (Figure 8C,D). In these FRET spikes, the SSB is pushed toward the 3′ ssDNA end by translocases (5′-3′), followed by displacement from the DNA end. It is noteworthy that despite the high affinity of *E. coli* SSB for ssDNA, *E. coli* SSB tetramers are dynamic on ssDNA, capable of diffusing along ssDNA at 1D diffusion coefficients, D1 = 270 nt^2^/s at 37 °C [33,63,67], and undergoing direct intra- or intersegment transfer to other target DNA sites [220,221,222,223]. A possible mechanism for reorganizing and removing tightly bound SSBs from ssDNA by translocases is presented by this single-molecule study that translocase can chemically and mechanically push heterologous SSB proteins along ssDNA. In this process, the SSB is pushed along the ssDNA in the direction of the desired translocation using nonspecific chemo-mechanical pressure [220].

A brief summary of the consequences of SSBs encountered by a translocating protein can be classified into four categories, determined by the characteristic of SSB and the translocating protein, their mutual interaction, and the given tasks. Firstly, SSBs bound on ssDNA may be dislodged by a protein translocating on the same strand. Secondly, the SSBs can stay on ssDNA and be bypassed without dissociation in the case of highly dynamic translocating proteins. Further, the SSB binding mode to ssDNA may be modulated to make space for the protein partner. Lastly, a diffusive SSB may be pushed along ssDNA through chemo-mechanical interaction.

**Figure 8 ijms-24-02806-f008:**
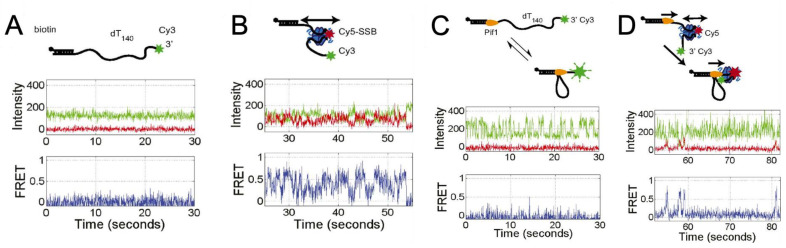
Example of single-molecule FRET assay to chemo-mechanical pushing of *E. coli* SSB by translocating protein partner Pif1 [220]. (**A**) A representative fluorescence trace of a Cy3–ssDNA (green) construct displays only Cy3 fluorescence without SSB or Pif1. (**B**) Following the binding of Cy5–SSB (red) to ssDNA constructs, fluctuating FRET signals are observed as a result of SSB diffusion along ssDNA. (**C**) Adding Pif1 (and ATP as an energy source) to the Cy3–ssDNA construct generates repetitive Cy3 enhancement spikes (PIFE) in the absence of SSB. (**D**) After Cy5–SSB is prebound on Cy3–ssDNA, Pif1 is then added with ATP into the experimental solution, replacing the FRET signal of SSB diffusion with the PIFE signal of Pif1 translocation. Periodically occurring asymmetric FRET spikes indicate SSB is pushed by Pif1 in a 5′ to 3′ direction. Images reused from [220]. Copyright 2016 National Academy of Sciences.

## 6. Conclusions

### 6.1. General Remarks on SSB

As transient and exclusive binders to ssDNA intermediates, SSBs are crucial for genome maintenance. Further interacting with various proteins vital for DNA maintenance, the SSB protein bridges genome maintenance pathways and modulates their activity by recruiting them to their DNA sites of action. Although it has been demonstrated that SSB is essential for DNA metabolism, the dynamics of SSB interaction with DNA and its partners remain unclear. Recent developed single-molecule assays have provided essential insights into the molecular mechanisms of SSB and interacting dynamics with other protein partners. By combining robust ensemble biochemical techniques and structural methods, a more comprehensive understanding of these SSBs has been gained.

This review summarizes the insights gained from single-molecule studies, which suggest that SSBs from various organisms show significant similarities. The OB binding domain from SSBs binds with high affinity to ssDNA, and the C-terminal tail of an SSB plays a crucial role in regulating the other protein partner activity. After binding and acting on ssDNA, these SSB–ssDNA complexes need to be bypassed, dislodged, pushed, or reorganized along the ssDNA to complete replication, recombination, and repair in a C-terminal tail-dependent manner or direct chemo-mechanical pushing fashion. As we have attempted to emphasize in this review, the SSB protein becomes a central scaffolding protein, rather than an accessory player, which contributes significantly to the storage and reliability of genomic information. In addition to defining the substrates upon which DNA replication, recombination, and repair must occur, SSB proteins function actively in nucleating enzyme complexes that are crucial to genome biology.

### 6.2. Potential Interesting Single-Molecule Experiments of SSBs

Taking a broader view, the single-molecule techniques discussed in this review are expected to significantly contribute to future research on SSBs in combination with biochemical ensemble techniques and structural tools. While significant progress has been made over the past 20 years, there is still much room for further research into SSB proteins’ molecular mechanisms. Here, the authors attempt to suggest several potential directions for studies of single molecules of SSBs.

#### 6.2.1. How the DNAp Displaces SSB from Different Organisms

Recent technical developments have allowed for visualization of the pathways of a diffusive *E. coli* SSB after encountering a translocative protein partner. Considering the fact that SSBs from various organisms are differentiated in terms of diffusion properties, it will be fascinating to track the fate of other static SSBs after encountering a moving partner, as we summarized in Section 5.5. An example case here can track the interplay of replicative polymerase encountering a static SSB on the lagging strand DNA.

#### 6.2.2. Hybrid SSBs from Both Host and Viral Organisms Interacting with ssDNA

Visualizing the interaction of viral SSBs and host *E. coli* SSBs on a stretched ssDNA template, for example, by combining dual-optical tweezers with confocal microscopy, will allow us to gain a better understanding of the binding dynamics and interplay of a hybrid SSB system to mimic a replisome in a viral infected host cell.

The context of replication makes for a very interesting system, and this ties into questions regarding co-evolution, not merely of residues within a protein or even inter-protein interactions, but also that of co-evolution between species, a special ‘intimate’ case of which is host–virus interactions.

#### 6.2.3. SSB Functions within a Complete Replisome

In addition to some recent attempts [224,225,226,227,228,229], further investigation into the interactions of SSBs with their homologous helicases and DNA polymerases to better understand replication would greatly benefit our knowledge of the complete replisome. Using advanced tools such as well-controlled mechanical manipulation and high-resolution imaging could aid in uncovering the intricate and regulated series of steps involved in the replication. The need for tight regulation and precise spatiotemporal coordination highlights the importance of understanding the role of SSBs in DNA replication.

#### 6.2.4. SSB as a Drug Target

Further research is needed to determine how protein complexes with SSB are controlled in vivo. This will help to identify which of the many competing interactions will prevail in a given situation. Additionally, the c-terminal tail of SSB may provide a unique feature against which new antibacterial therapies can be developed.

## Figures and Tables

**Table 1 ijms-24-02806-t001:** Example output of single-molecule studies.

Phenomenon	Quantities	Structural Insights	Example Study
Binding kinetics	Time constants of binding (k_on_ and k_off_)	Binding steps, timescales of binding processes	[76]
Binding footprint	Binding footprint from density	DNA binding pocket	[77]
Binding thermodynamics	Differential stability is based on temperature or applied force. Possible to calculate by FEC hysteresis	Binding stability and reaction energetics	[2,77]
Diffusion	Diffusion constant, velocity, direction	Directionality of movement, interaction with DNA (wrapping, base interference, etc.)	[33]
Cooperativity	Cooperativity score, based on concentration-dependent binding affinities	Interactions between SSB units	[22,78]

## Data Availability

It should be noted that all data, including figures presented in this review, are derived from previously published data sets, as indicated in the figure captions. We have superimposed the axes lines and tick marks directly over the original markings to ensure consistency in line thickness. Additionally, some figures have been annotated with arrows, text, and lines to highlight specific relevant characteristics in the data.

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
