# Peer review of "Unravelling How Single-Stranded DNA Binding Protein Coordinates DNA Metabolism Using Single-Molecule Approaches"

_ijms, 2023, doi:10.3390/ijms24032806_

Round 1
Reviewer 1 Report
Comments:
1. The section “Properties of SSBs” should be more elaborated for the in-depth knowledge of the overall readers.
2. Clarity of the figures is one of the major issues in this article. E.g., the resolution of Fig 1, Fig 3, Fig 6, etc. must be enhanced for clear visualization.
3. The authors discussed smFRET techniques for the movement of SSBs proteins. Is the FCS technique not useful in this case? Please check other reports where the movements were measured by the FCS. If not, please provide a short discussion on this issue.
4. In smFRET data analysis, which models were used to fit the data? Discuss this part. The model of fitting can tell us the in-depth story of the dynamics.
5. Most of the references are old compared to advanced studies. Please discuss more recent studies and include them in citations. The reference section should contain new sources to reveal the relevance of the work.
6. The advantage of this study in applications of the new era of science should be discussed in the article.
8. Some typos are present. E.g., In Page 4, the “Kd” should be replaced with “kd”. These should be modified.
Reviewer 2 Report
This manuscript reviews the scientific literature concerning the ways in which single-stranded DNA binding proteins play a role in DNA replication and repair processes, with a focus on single molecule studies. The review is generally well-written and relatively easy to follow. The selections of figures support the discussion in the text and the content is divided into logical sections producing a review that I feel usefully collects and summarises the literature in this area of research. There are some points that I feel it would be useful for the authors to address and these are detailed below.
1. Title. I am not comfortable with the term “DNA transaction” as essentially it is a non-standard and non-specific term which I assume the authors are intending to use as a catch-all term for replication and repair processes. The authors should consider rewording this.
2. Page 2 lines 55-56 “. . . in a C-terminal . . . chemo-mechanical pushing fashion.” I don’t think that these two modes of action have been well-explained at this point in the manuscript. I think it would be worthwhile adding a brief elaboration of the meaning of these mechanisms to improve readability.
3. Page 4 line 143 “. . . pulling rate . . .” I think this term ought to be succinctly defined at this point in the manuscript.
4. Page 6 line 246 Change “. . . binding constants . . .” to “. . . binding rate constants . . .”
5. Page 7 lines 292-293 “. . . the one-dimensional parking problem . . .” I think this term ought to be succinctly defined at this point in the manuscript.
6. Page 13 lines 500 and 508 “. . . red (i) . . . green (iv) . . .” and “. . . i and ii, and iv” These passages lack reference to panel iii in the figure and therefore need to be corrected.
7. Page 14 line 576 Replace “. . . with 10-20-fold . . .” with “. . . by 10-20-fold . . .”
8. Page 15 line 619 “. . . of Cy5 without no recovery, . . .” This is clumsy construction and does not appear to reflect what you mean. Modify this passage of the text.
9. Define the abbreviation XPD in the text of the Figure 5 legend. All figures should be comprehensible independently of the main text.
10. Page 18 lines 742-743 It is not clear what you mean by “correcting the forward movement of the helicase”. Clarify this in the manuscript.
11. Page 19 lines 780-781 Clarify what an “As am secondary structure” is in this context.
12. The reader is not left with a clear impression of where research in this field might head in the future. Either expand your conclusions section or add another section to address the possible future directions for the research based on your assessment of the current literature.
Reviewer 3 Report
The topic is interesting indeed, however there are several issues about this manuscript. First of all, it is rather long and hence difficult to follow; Moreover, such length is unnecessary because the authors repeat the same ideas over and over again. They give too much weight to the new methodologies with single-molecule approach and neglect the truly interesting part, the similarities or differences among the SSB proteins, seems like they use the SSB proteins to analyze the methodologies and not the opposite. An excellent example of this is the part where they describe the interaction of SSB with RecA, they merely mention the SOS response induction and forget all about the possible role of the RecFOR complex in this process.
Round 2
Reviewer 3 Report
The manuscript is suitable for publication